# Shear Stress-Induced Activation of von Willebrand Factor and Cardiovascular Pathology

**DOI:** 10.3390/ijms21207804

**Published:** 2020-10-21

**Authors:** Sergey Okhota, Ivan Melnikov, Yuliya Avtaeva, Sergey Kozlov, Zufar Gabbasov

**Affiliations:** 1National Medical Research Centre of Cardiology of the Ministry of Health of the Russian Federation, 15A, 3-rd Cherepkovskaya Street, 121552 Moscow, Russia; ae2007@mail.ru (S.O.); ivsgml@gmail.com (I.M.); julia_94fs@mail.ru (Y.A.); bestofall@inbox.ru (S.K.); 2State Research Center of the Russian Federation—Institute of Biomedical Problems of Russian Academy of Sciences, 76A, Khoroshevskoye Shosse, 123007 Moscow, Russia

**Keywords:** von Willebrand factor, ADAMTS-13, atherosclerosis, atherothrombosis, coronary artery disease, Heyde’s syndrome

## Abstract

The von Willebrand factor (vWF) is a plasma protein that mediates platelet adhesion and leukocyte recruitment to vascular injury sites and carries coagulation factor VIII, a building block of the intrinsic pathway of coagulation. The presence of ultra-large multimers of vWF in the bloodstream is associated with spontaneous thrombosis, whereas its deficiency leads to bleeding. In cardiovascular pathology, the progression of the heart valve disease results in vWF deficiency and cryptogenic gastrointestinal bleeding. The association between higher plasma levels of vWF and thrombotic complications of coronary artery disease was described. Of note, it is not the plasma levels that are crucial for vWF hemostatic activity, but vWF activation, triggered by a rise in shear rates. vWF becomes highly reactive with platelets upon unfolding into a stretched conformation, at shear rates above the critical value (more than 5000 s^−1^), which might occur at sites of arterial stenosis and injury. The activation of vWF and its counterbalance by ADAMTS-13, the vWF-cleaving protease, might contribute to complications of cardiovascular diseases. In this review, we discuss vWF involvement in complications of cardiovascular diseases and possible diagnostic and treatment approaches.

## 1. Introduction

The von Willebrand Factor (vWF) is a large multimeric glycoprotein, present in blood plasma, endothelial cells, megakaryocytes, and platelets. It plays a major role in hemostasis, mediating platelet adhesion to vascular injury sites. It also binds and protects coagulation factor VIII (FVIII) from degradation [1]. Recent studies showed that vWF is also involved in inflammation, linking thrombosis and inflammation. Inflammation can provoke thrombosis through the vWF-dependent pathway, which includes endothelial activation, the secretion of vWF into the bloodstream, vWF activation and interaction with platelets, and subsequent platelet adhesion to a vessel wall [2,3,4]. vWF multimers and platelets, adhered to injured and activated endothelium, might serve as sites of leukocyte recruitment. Altogether, this predisposes to the propagation of the inflammatory process. vWF is considered a risk factor of arterial thrombosis and might contribute to the development of adverse events in atherosclerosis and other cardiovascular diseases [5,6,7,8].

## 2. Structure and Functions of VWF in Bloodstream

vWF is mainly produced in endothelial cells and stored in Weibel–Palade bodies—the storage granules of endothelium, consisting of densely packed vWF multimers and P-selectin [9]. A small amount of total vWF is produced in megakaryocytes and stored in the α-granules of platelets. The mature vWF monomeric molecules form dimers through C-terminal disulphide bonds. Dimers compose the basic vWF repeating structure, with a molecular mass of approximately 500 kDa [10]. Dimers are then polymerized into ultra-large multimers through N-terminal disulphide bonds, with the size of stored multimers ranging from 20 to 100 dimers [11].

vWF is constantly secreted from the Weibel–Palade bodies of endothelial cells into the bloodstream, and from the α-granules of platelets upon activation. The pool of circulating vWF consists of multimers of various sizes, ranging from a few dimers to high-molecular weight multimers (HMWM), which contain 11–20 dimers. The former basically serve as FVIII carriers, while the latter present the main hemostatically active force of vWF [11]. The more dimers that are in a vWF molecule, the more hemostatically active it is. Upon secretion, large vWF multimers come into contact with metalloproteinase ADAMTS-13, the vWF-cleaving protease. ADAMTS-13 regulates the size of circulating vWF through proteolytic cleavage of its multimers [12].

vWF contains a variety of domains, which define the characteristics of the molecule: A1, A2, A3; D assemblies: D1, D2, D’D3, D4; and the C-terminal cysteine knot (CTCK) (Figure 1) [13]. The A1 domain binds mainly glycoprotein (GP) Ib receptors of platelets (the only receptor on non-activated platelets that has pronounced affinity to vWF), and to a lesser extent, collagen types I, IV, VI, and heparin. The A2 domain presents a binding cite for ADAMTS-13. The main collagen binding sites are located on the A3 domain. The C1 domain can interact with activated platelets through the binding of the IIb/IIIa receptors. The D’D3 assembly carries FVIII. The CTCK domain dimerizes vWF. All D assemblies are involved in the assembly and disulfide linkage of vWF dimers into long tubules of the Weibel–Palade bodies. [9,13]. vWF binds leukocytes through the interaction of the A1 domain with P-selectin glycoprotein ligand-1 (PSGL-1), the D’D3, and the A1, A2, and A3 domains with β2-integrins [14].

vWF exists in the bloodstream in one of two conformations—globular and unfolded [15]. vWF conformation depends on the shear rate of blood flow in vessels. The shear rate is the rate of change of velocity, at which one layer of fluid passes over another, and is measured in inverse, or reciprocal, seconds (s^−1^). This parameter is used to estimate flow of liquids in tubes, which is important for the understanding of vWF activation. Under conditions of low shear rates (e.g., in veins, where shear rates are 15–200 s^−1^, or in large arteries, where they are 300–800 s^−1^), vWF remains in a globular shape, which hides the binding sites, and does not interact with the circulating platelets [16]. In the case of vessel injury and under high shear rates (e.g., in intact small arteries and arterioles, shear rates are 450–1600 s^−1^, but at sites of advanced atherosclerosis, shear rates might reach up to 11,000 s^−1^ and even higher), the vWF shape unfolds, and opens binding sites, in particular for glycoprotein (GP) Ib receptors of platelets.

A threshold value of shear rate, critical for unfolding and the activation of vWF, is estimated at 5000 s^−1^ [15]. Uniform shear rates occur in straight channels, but in pathological conditions in stenotic or injured vessels, elongational flows occur, where the molecules are subjected to shear gradients. The vasoconstriction of injured vessels and stenoses of arterial lumen by atherosclerotic plaques create two zones of elongational flow. Flow and shear rate accelerate at the inflow and decelerate at the outflow of a lesion. At the inflow, elongation occurs parallel to the flow direction. At the outflow, elongation occurs perpendicular to the flow direction because the streamlines diverge. As the adjacent shear layers have different velocities, large molecules, such as vWF multimers, are subjected to the rotational and elongational components of the flow. In the straight sections of a blood vessel, the rotational and elongational components are balanced. However, at the inflow and the outflow of a lesion, the equilibrium is disturbed. At sites of vasoconstriction and stenoses, thrombus formation is observed at the outflow zone of a lesion (Figure 2) [9]. Elongation flows are predicted to unfold vWF at rates of two orders of magnitude below the corresponding pure shear rate values [17]. vWF-dependent thrombus formation was observed at the outflow region of an in vitro model of stenosis at the inflow shear rates of 600, 1000, and 2000 s^−1^ [18].

Unfolded by high shear rates, the vWF threads can self-associate and form long strands and web-like structures, further stimulating platelet adhesion [19]. The ability of unfolded vWF to self-associate under high-shear rates was studied in vitro in endothelialized microvessels. Smaller vessels, turns, bifurcations, flow acceleration predisposed to thickening, and elongation of vWF strands. In regions with complex flow, the vWF strands tended to form web-like structures. At the same time, at sites of low shear rates, the vWF remained in a globular conformation and no vWF strands were visible [19]. Schneider et al. also reported spider web-like network formation by vWF, under high shear rates [15]. Zhang et al., demonstrated that vWF association was dependent on the A2 domain under shear rate of 9600 s^−1^. Hence, the A2 domain might have an overlapping role, both in proteolysis and self-association of vWF [20].

The role of platelet vWF in hemostasis is not clearly established, and the studies here are quite sparse. A study on the vWF-knockout mice showed that replenishing of vWF in platelets after bone marrow transplantation from wild-type mice, resulted in partial correction of bleeding time and reduction of blood loss volume [21]. Recently, a study showed that chimeric mice with only platelet vWF had a bleeding time comparable to that of the vWF-knockout mice [22]. Platelet vWF did not contribute to thrombus formation in a model of carotid artery injury. On the other hand, after ischemic brain injury, the mice with only platelet vWF and without plasma vWF, had a cerebral infarction size similar to the wild-type mice. Platelet vWF mediated ischemic brain injury in a GPIb-dependent mechanism and significantly contributed to intracerebral thrombosis formation [22].

Upon the unfolding of vWF multimers, FVIII is exposed on D’D3 domains [1]. FVIII participates immediately in the intrinsic coagulation pathway, interacting with factor IXa as a cofactor to form the intrinsic tenase (enzyme complex that cleaves inactive coagulation factor X into active Xa). FVIII is essential for normal hemostasis—its deficiency leads to hemophilia A. vWF acts as a carrier of FVIII in circulation. In the absence of vWF, FVIII is unstable and exposed to rapid degradation. vWF significantly elongates its half-life and protects FVIII from proteolysis, eventually delivering it to sites of vessel injury [1].

Unraveling, vWF faces its antagonist, ADAMTS-13 [9]. It is a member of the ADAMTS (A Disintegrin And Metalloprotease with ThromboSpondin type 1 repeats) family of proteolytic enzymes. ADAMTS-13 was first isolated in 2001, though the presence of Ca^2+^-dependent vWF-cleaving protease was predicted earlier [23]. Human ADAMTS-13 mRNA and protein synthesis is localized exclusively to hepatic stellate cells [24]. The plasma levels of ADAMTS-13 show negative correlation with the plasma levels of vWF [25]. In the absence of vWF in the bloodstream, i.e., in patients with von Willebrand disease (vWD) type 3, the plasma levels of ADAMTS-13 are 30% higher than in healthy volunteers. On the contrary, after the administration of 1-deamino-8-D-arginine vasopressin (DDAVP), which stimulates the release of vWF from endothelial Weibel–Palade bodies, the plasma levels of ADAMTS-13 showed a 20% reduction in healthy volunteers [25]. The negative correlation is likely to occur because of consumption in the cleavage of vWF. ADAMTS-13 circulates in the bloodstream as an active enzyme, but vWF in the globular conformation remains resistant to proteolysis. Once vWF unfolds, it exposes the binding sites for ADAMTS-13 on the A2 domain. The ensuing interaction with ADAMTS-13 leads to the cleavage of vWF multimers between tyrosine in the 1605 position and methionine in the 1606 position, which shortens the multimers and decreases the hemostatic activity of vWF [16,26].

The growing evidence shows that vWF is an important mediator of vascular inflammation [27]. A murine model of acute peritonitis showed that vWF plays an important role in leukocyte recruitment and extravasation into the site of inflammation. Notably, leukocyte extravasation requires vWF-platelet binding, as inhibition of the platelet GPIb receptors prevented vWF-mediated leukocyte extravasation [28]. A murine model of immune complex-mediated vasculitis and irritant contact dermatitis showed that the interference with vWF activity with vWF-blocking antibody led to a substantial decrease in vWF-mediated leukocyte recruitment and reduction of cutaneous inflammatory response. In this study, the vWF-blocking antibody did not interfere with the GPIb domain [29]. Another murine model with the same pathology but with the use of the antibody targeting the A1 domain showed comparable results [30]. A study comparing features of acute inflammation after focal cerebral ischemia in vWF-deficient and ADAMTS-13-deficient mice, showed that the vWF-deficient mice had a smaller injury size, reduced cytokine release and neutrophil infiltration of the injury site, compared to the ADAMTS-13-deficient mice [31]. A myocardial ischemia/reperfusion study in ADAMTS-13-knockout mice showed that infusion of recombinant ADAMTS-13 substantially reduced myocardial apoptosis, troponin-I release, and resulted in a 9-fold reduction in the number of neutrophils infiltrating the ischemic zone [32].

## 3. Diagnostic Tests for the von Willebrand Factor Deficiency and Dysfunction

Studies of acquired von Willebrand syndrome in aortic stenosis and obstructive hypertrophic cardiomyopathy revealed that, despite the clinical manifestation of bleeding, plasma levels of vWF and ristocetin cofactor assay remained at normal values [33]. The plasma levels of vWF show mass concentration of all vWF multimers, from dimers to HMWM. The proteolyzed HMWM of vWF are cleaved into smaller multimers and dimers. This significantly affects the hemostatic function of vWF, but not its mass concentration. Thus, measuring the plasma levels of vWF shows the amount of protein in the plasma, but not its functional status. The plasma levels of vWF depend on the blood group of a person [34]. Variations of the plasma levels of vWF between blood groups might complicate distinguishing healthy persons with low vWF from mild cases of vWD [35]. Unlike oligosaccharides of other plasma glycoproteins, the vWF structure includes oligosaccharides of the ABO blood groups. Persons in the O blood group have significantly lower levels of vWF than persons in other blood groups [36]. A study, comprising 1117 blood donors showed that the plasma levels of vWF were the lowest in the O blood group (74.8 IU/dL), higher in the A blood group (105.9 IU/dL), even higher in the B blood group (116.9 IU/dL), and the highest in the AB blood group (123.3 IU/dL) [37]. In ristocetin cofactor assay, ristocetin was used to bind the GPIb receptors of platelets to the A1 domain of vWF [38]. The assay was performed under very low shear rate conditions, when the vWF multimers remained folded. Thus, the assay reflected the presence of vWF in a sample per se, but not the physiological hemostatic function of vWF. It also had a low sensitivity to the loss of HMWM of vWF. Hence, it was useful in the diagnosis of vWD, especially in the case of severe vWF deficiency (e.g., in type 3 vWD, in which vWF was almost or completely absent in the blood), but provided little diagnostic information concerning the actual hemostatic function of vWF. Significant interlaboratory variations in test results existed for both of these assays [38]. vWF multimer analysis by the electrophoresis of vWF on agarose gels remained the main assay of qualitative deficiency of vWF [10]. The assay had excellent sensitivity for the loss of HMWM of vWF. The inclusion of the luminescent methods gave it the power to detect the presence of vWF in samples with a concentration of less than 1 IU/dL. This allowed differentiation between homozygous type 2 and type 3 vWD. vWF multimer analysis is crucially important for subtyping of type 2 vWD [10]. It is also very useful in diagnostic of HMWM of vWF deficiency in acquired von Willebrand syndrome, due to the heart valve disease [39]. However, the assay placed high demands on the technical competence of the laboratory personnel and was unsuitable for standardization [35]. Therefore, its introduction to routine diagnostics in cardiovascular disease is challenging. vWF collagen binding assay might be used as a substitution for the detection of HMWM of vWF [40]. Thus, the broad panel of tests is needed to diagnose vWD and its type. To simplify testing (e.g., in screening for bleeding disorders or in the emergency setting), a screening tool that can quickly exclude platelet defects and vWD was introduced. The Platelet Function Analyzer-100 (PFA-100) is an easy to use device, which utilizes whole blood and requires only 5 min to complete a test [41]. Currently, this is the only commercially available system that evaluates primary hemostasis under high shear rate conditions. PFA-100 pumps a whole blood sample through a narrow orifice in a membrane, covered with collagen and platelet activator agents, such as adenosine diphosphate (ADP) or epinephrine. The system measures time from the start of a sample pumping to the closure of the orifice (the closure time). The shear rates in the orifice might reach 5000–6000 s^−1^, which is sufficient to activate vWF. However, PFA-100 possesses a drawback concerning vWF activation. The system is designed to assess primary hemostasis, not exclusively vWF hemostatic function. The closure time is dependent on interactions between platelets and other blood cells, not vWF activation alone. Its sensitivity varies, depending on the severity of vWF deficiency and platelet defects, from moderate in mild cases to 100% sensitivity in the case of complete absence of vWF in blood [41,42]. Thus, currently there is no widely available assay that exclusively measures vWF shear rate-dependent hemostatic function. Experimental assays and devices, such as microfluidic chambers, are created by some research laboratories to study different aspects of hemostasis, including vWF functions. However, these devices are cumbersome and not standardized. Therefore, currently, these devices fall short of satisfying clinical demands [43].

## 4. Diseases, Associated with von Willebrand Factor and ADAMTS-13 Dysfunction

vWF plays a major role in coagulation, and its deficiency or dysfunction predisposes one to bleeding. A genetic disorder, caused by partial or total deficiency or structural aberrations of vWF molecules, is called vWD [35]. This is one of the most widespread hemostasis disorders with an incidence of around 1:100 persons. It is characterized by nasal and gingival bleeding, hemarthrosis, muscular and subcutaneous hematomas, menorrhagia, and prolonged bleeding in traumas. Desmopressin, which stimulates the release of vWF from endothelium, is the main medication for the mild forms of vWD. In severe cases, the only choice is regular vWF-rich plasma transfusions [11]. Acquired von Willebrand syndrome is a rare disorder that occurs due to lymphoproliferative (e.g., chronic lymphocytic leukemia), myeloproliferative (e.g., thrombocythemia), cardiovascular (e.g., aortic stenosis) and immunological disorders (e.g., hypothyroidism), which provoke the qualitative and quantitative deficiency of vWF [44].

On the contrary, increased concentrations of ultra-large multimers of vWF lead to thrombotic thrombocytopenic purpura (TTP), which develops due to a deficiency of ADAMTS-13 [45]. ADAMTS-13 deficiency causes an unrestricted presence of ultra-large multimers of vWF in the bloodstream, which might spontaneously interact with platelets and provoke thrombosis in multiple small vessels. This results in thrombocytopenia with purpura, hemolytic anemia, and multiple organ damage, due to microvascular thrombosis—acute renal, heart and neurological damage, mental derangement, and fever. TTP can develop either due to genetic anomalies (congenital TTP) or production of autoantibodies to ADAMTS-13 (acquired TTP), which interfere with its function or enhance clearance. The determination of the ADAMTS-13 function is essential in TTP diagnosis. TTP usually manifests when the ADAMTS-13 levels fall below 10%, which is below the detection limit of many assays. As soon as ADAMTS-13 activity becomes detectable (i.e., higher than 10%), TTP symptoms resolve. Differentiation between congenital and acquired TTP is vitally important, as treatment of these life-threatening conditions significantly differ. The diagnosis of congenital TTP is built on the detection of pronounced fall in ADAMTS-13 activity, ruling out autoantibodies to ADAMTS-13, and analyzing the ADAMTS-13 gene. The diagnosis of acquired TTP requires the lack of ADAMTS-13 activity and the detection of autoantibodies to ADAMTS-13. Patients with congenital TTP respond well to the plasma exchange therapy, which improves survival rates from 10% to 80–90%. During the plasma exchange, ultra-large multimers of vWF, immune complexes, and autoantibodies are removed. Infusions of donor plasma with normal content of ADAMTS-13 are also effective, and used in combination with the plasma exchange therapy. Patients with acquired TTP respond to normal plasma infusions poorly. Suppression of autoantibody production with corticosteroids, destruction of CD20-posititve B-lymphocytes with rituximab, or the use of other immunosuppressive agents is considered to be effective in acquired TTP. Splenectomy is effective, but reserved for refractory cases of acquired TTP [46]. Recently, caplacizumab, an immunoglobulin fragment that targets the A1 domain of vWF, and prevents its interaction with platelet GPIb receptors, was introduced in addition to plasma exchange therapy [47].

## 5. Acquired von Willebrand Syndrome in Heart Valve Disease and Hypertrophic Cardiomyopathy

The association between aortic stenosis and cryptogenic gastrointestinal bleeding was first described in 1958 in a letter by E.C. Heyde [48]. In 1992, Warkentin et al. were the first to suggest a link between the loss of HMWM of vWF that develops due to aortic stenosis or hypertrophic cardiomyopathy and bleeding from gastrointestinal angiodysplasia [49]. In 2002, Warkentin et al., was also the first to provide a rationale for this association [50]. They reported two cases of severe aortic stenosis with concomitant gastrointestinal bleeding that ceased after successful replacement of the aortic valve. Notably, prior to the platelet count operation, activated partial-thromboplastin time, the plasma level of FVIII, the plasma levels of vWF, and the ristocetin cofactor assay values were normal, whereas the loss of HMWM of vWF was severe. After the valve replacement, the percentage of HMWM recovered and remained normal during a 10-year follow up [50]. Acquired von Willebrand syndrome in cardiovascular disease develops due to aortic stenosis, left ventricle outflow tract obstruction in hypertrophic cardiomyopathy, and aortic and mitral regurgitation [8]. In severe aortic stenosis, a steep increase in shear rates in the aortic orifice and left ventricular outflow tract occur. HMWM of vWF, which pass through the narrowing aortic valve, undergo activation and subsequent proteolysis by ADAMTS-13. Additionally, lower pulse pressure, which is observed in aortic stenosis, contributes to the decreased secretion of HMWM of vWF from the endothelium [8]. In severe aortic stenosis, HMWM of vWF might decline to approximately 1/2 of the normal levels [51]. Panzer et al., reported a study of 47 patients with severe aortic stenosis, who were subjected to aortic valve replacement. HMWM of vWF were depleted in all patients before valve replacement and became normal in most patients. PFA-100 closure time in collagen-ADP cartridges were significantly prolonged at the baseline, and normalized after the treatment. Using the cone and plate analyzer, the authors showed that the loss of HMWM of vWF affects platelet adhesion and, to a larger extent, ADP-induced platelet aggregation [52]. The quantitative deficiency of HMWM of vWF leads to the acquired von Willebrand syndrome type 2A. Its clinical manifestation in patients with heart valve disease is called Heyde’s syndrome [53]. It is characterized by cryptogenic gastrointestinal bleedings from submucous arterial malformations, developing together with the progression of aortic stenosis and its cardiovascular symptoms (such as fatigue and angina). Bleeding from arterial malformations is thought to be provoked by increased shear rates, due to the complex structure of vessels, which cause further consumption of vWF [53]. Patients with severe aortic stenosis also report skin and mucosal bleeding. In a study of 50 patients with aortic stenosis, skin or mucosal bleeding was reported by 21% of participants [33]. Primary hemostasis, measured by PFA-100 was significantly prolonged; the percentage of HMWM and the collagen-binding activity of vWF were reduced in patients with severe aortic stenosis. The plasma levels of vWF were normal in all patients. One day after surgical treatment, PFA-100 values and the percentage of HMWM of vWF were completely corrected; however, after 6 months, these values deteriorated, primarily in patients with prosthesis mismatch [33]. In two other studies of 183 and 21 patients with severe aortic stenosis, undergoing valve replacement, the same pattern was observed. PFA-100 closure time was prolonged and the HMWM of vWF reduced at the baseline; they normalized after successful valve replacement [51,54]. The volume of blood that is affected by high shear rates is a crucial component in the development of Heyde’s syndrome, in which the whole volume of circulating blood is affected. For example, in patients with severe coronary or peripheral atherosclerosis, vWF deficiency does not occur because blood volume is affected only partially. The treatment of aortic stenosis with valve replacement results in fast recovery and the cessation of bleeding. The normalization of shear rates and pulse pressure leads to an increase in the plasma level of functional HMWM of vWF within hours [55]. A prospective study with an 18-month follow-up of patients treated with aortic valve replacement, and several other studies with follow-up from 2 weeks up to 6 months, showed that, after recovery, the plasma levels of HMWM of vWF did not drop again [51]. However, if significant aortic regurgitation and paravalvular leak develop after aortic valve replacement, the plasma levels of HMWM of vWF might not recover [54]. A similar development pathway of the acquired von Willebrand syndrome was also reported in patients with aortic or mitral valve regurgitation [56,57].

In obstructive hypertrophic cardiomyopathy, the obstructed left ventricle outflow tract predisposes the proteolysis of HMWM of vWF in a manner similar to aortic stenosis. A study by Blackshear et al., included five patients with symptomatic obstructive hypertrophic cardiomyopathy [58]. Spontaneous gastrointestinal, mucosal, or excessive postsurgical bleeding was observed in all patients. The plasma levels of vWF and ristocetin cofactor assay values were within normal limits, whereas electrophoresis revealed loss of HMWM and excess of low-molecular weight multimers of vWF. After surgical septal myectomy, bleeding resolved, and HMWM were restored to normal levels in all patients [58]. In another study of 28 patients with obstructive hypertrophic cardiomyopathy, plasma levels of vWF were normal in all patients [59]. PFA-100 closure time was significantly prolonged in all but one patient. Additionally, all patients had a substantial reduction in HMWM of vWF. A strong correlation of closure time in the PFA-100 tests and reduction in HMWM of vWF with peak gradients, measured by ultrasonography, was observed. The mean peak gradient, sufficient for the impairment of vWF function and the reduction in HMWM, was estimated at 15 mm Hg [59].

The use of left ventricle assist devices (LVADs) might also lead to acquired von Willebrand syndrome [60]. LVADs provide circulatory support for patients with end-stage heart failure, either as a bridge or destination therapy. vWF might immobilize on the biomaterial surfaces of LVAD and undergo cleavage at sites of high shear rates, which might occur in the LVAD system. This might lead to the development of LVAD-associated complications, such as pump thrombosis and bleeding; in particular, gastrointestinal bleeding from arteriovenous malformations [60].

## 6. von Willebrand Factor and Coronary Artery Disease

The relevance of vWF in platelet adhesion and thrombus formation under high shear rates was shown in a study on pigs [61]. In this study, 8 healthy pigs and 6 pigs with vWD were fed a high-cholesterol diet for 24 weeks with the assessment of coronary arteries for the presence of atherosclerosis. The diet led to severe hypercholesterolemia (7–39 mMol/L). Coronary atherosclerosis developed in both groups and was detected by histology in all but one healthy and all vWD pigs. Left anterior descending coronary and carotid arteries were clamped to produce stenotic segments, which were then injured. Occlusive thrombosis occurred in all stenotic segments in phenotypically normal pigs, while it did not occur in the vWD pigs [61].

A study on mice showed the influence of ADAMTS-13 and vWF deficiency on a myocardial infarction (MI), induced by the ligation of the left anterior descending coronary artery [62]. The infarct size was significantly larger in homozygous ADAMTS-13-deficient mice (22.2 ± 1.1%), compared to heterozygous ADAMTS-13-deficient mice (17.3 ± 0.8%) and wild-type mice (16.9 ± 1.2%), which suggests that a level of approximately 50% of ADAMTS-13 in plasma is sufficient to prevent aggravated MI. The infarct size was markedly reduced in the vWF-deficient mice (7.3 ± 0.7%), compared to the wild-type mice (18.6 ± 1.3%). A group comprising ADAMTS-13 deficient and wild-type mice was treated with a polyclonal antibody to vWF. In this group, the infarct size in the wild-type and in the ADAMTS-13-deficient mice (8.5 ± 0.7% and 8.2 ± 1.3%, respectively) was significantly smaller than in the control wild-type mice treated with a nonspecific antibody (17.48 ± 0.7%) [62].

CAD in patients with vWD was observed less frequently. In a register study comprising 7556 cases related to vWD and 19,918,970 cases unrelated to vWD, CAD was less common in patients with vWD (15.0%) than in patients without vWD (26.0%). After multivariable logistic regression analysis with adjustment for the main risk factors of CAD, the probability of CAD in patients with vWD remained lower than in patients without it (OR 0.85; 95% CI 0.79–0.92) [63].

The plasma levels of vWF differ in healthy persons and patients with CAD. In 110 patients with a mean age of 58 ± 20 years with CAD, the plasma level of vWF was 141.78 ± 20.53 IU/dL, whereas in a control group of healthy volunteers it was 111.95 ± 17.15 IU/dL [64]. A prospective multicenter study comprising 3043 patients with stable angina or previous MI, showed that the higher plasma levels of vWF correlated with an 8.5% increase in the rate of MI and sudden cardiac death [65]. A meta-analysis of the prospective Reykjavik study, comprising 1925 persons who had primary nonfatal MI or died of CAD during a follow-up (median 19.4 years), and 3616 controls, showed that the baseline plasma levels of vWF were higher in patients with CAD than in the control group [66]. The ENTIRE-TIMI 23 study, comprising 314 patients with ST-elevation myocardial infarction (STEMI) in whom the plasma levels of vWF were measured before and 48–72 h after fibrinolysis, showed that an increase in vWF levels after fibrinolysis was associated with the higher mortality and MI rate in 30 days (11.2% vs. 4.1%, respectively) [67]. In 123 patients who had MI before the age of 70, plasma levels of vWF were measured 3 months after MI. A 4.9-year follow up showed that higher concentrations of vWF were independently associated with recurrent MI and mortality [68].

A study, comprising 1026 patients with confirmed first STEMI and 652 healthy controls, showed that the plasma levels of vWF were nearly 1.5-fold higher in patients with STEMI, than in healthy controls (median 378.2 vs. 264.4 ng/mL, respectively). The plasma levels of ADAMTS-13 were lower in patients with STEMI, than in the healthy controls (median 90% vs. 97%, respectively) [69]. In another study, the plasma levels of vWF in 41 patient of mean age 68 ± 23 years, hospitalized within 72 h after the onset of MI, were significantly higher (2151 ± 97 mU/mL) than in 33 patients with stable angina and 90% narrowing of a major coronary artery (1445 ± 93 mU/mL), who had angina attack within 4 weeks before the study, or 30 patients with chest pain syndrome without hemodynamically significant stenosis or coronary spasm (1425 ± 76 mU/mL), in whom the vWF levels did not differ significantly [70]. On the other hand, the plasma levels of ADAMTS-13 were significantly lower in patients with acute MI (799 ± 29 mU/mL) than in patients with stable angina (996 ± 31 mU/mL) or patients without hemodynamically significant stenosis or vasospasm (967 ± 31 mU/mL). The enzymatic activity of ADAMTS-13 was also lower in patients with acute MI (768 ± 27 mU/mL) than in patients with stable angina (893 ± 27 mU/mL) or patients without hemodynamically significant stenosis or vasospasm (936 ± 29 mU/mL). [70]. In the GLAMIS study of 466 patients of mean age 55.1 ± 7.45 years with acute MI and 484 healthy controls, no correlation between ADAMTS-13 and vWF levels was found in the healthy controls [71]. On the other hand, the plasma levels of vWF correlated positively and the plasma levels of ADAMTS-13 correlated negatively with the risk of MI. After adjustment for the main cardiovascular risk factors, each 60 IU/dL increase in plasma levels of vWF was expected to raise the risk of MI by about 35%. Each 33% increase in the plasma levels of ADAMTS-13 was expected to decrease the risk of MI by about 27% [71]. The SMILE study, which included 650 men, 18–70-years-old, with stable CAD, who had an MI event at least 6 months before the study, and 646 healthy men, revealed no clear association between ADAMTS-13 and the vWF plasma levels, measured more than 6 months after the onset of MI [72]. There was also no significant difference between the ADAMTS-13 levels in men with stable CAD and the healthy controls (mean levels 101% and 100%, respectively). Neither were the plasma levels of vWF different between the groups (mean 138% in the CAD group and 135% in controls) [72]. The prospective PRIME study, comprising nearly 10,000 healthy men, among whom 296 developed CAD during the 5 years of follow-up (158 persons with MI and 142 with stable and unstable angina), showed that the baseline plasma levels of vWF were significantly higher in men who developed MI (129.2 ± 53.1 IU/dL) compared to the healthy controls (115.9 ± 41.8 IU/dL) [73]. The relative risk of MI development was 3.34 in patients with plasma levels of vWF in the 4th quartile, compared to 1.0 in persons with vWF plasma levels in the 1st quartile. Stable and unstable angina incidence did not correlate with the plasma levels of vWF [73].

A study on rats attempted to determine time span for vWF plasma level recovery after STEMI [74]. The study comprised 57 male rats subjected to ligation of the left anterior descending artery, 2 mm away from bifurcation, which provoked ECG-verified STEMI and myocardial fibrosis, on histological examination. Rats were randomized into four groups—in the first, blood was taken from the coronary sinus and the inferior vena cava, 1 h after the onset of MI; in the second, 24 h after MI; in the third, 7 days after MI; the fourth was used as a control. The plasma levels of vWF, obtained from the coronary sinus, increased 1.31-fold after 1 h, and 0.88-fold, 24 h later. They decreased to normal levels on the seventh day. In the inferior vena cava, the plasma levels of vWF were 0.37-fold higher 1 h after the onset of MI, 0.18-fold higher 24 h later, and decreased to normal levels on the seventh day [74].

## 7. Inflammatory and Stress Stimuli as a Possible Cause of Elevation in Plasma von Willebrand Factor in Coronary Artery Disease

There are several reasons for the increase in plasma levels of vWF in CAD and, in particular, MI. In a 5-year follow-up of 1592 persons 55–74 years old, smoking was associated with higher levels of vWF and a higher risk of atherosclerosis development [75]. In a study of 2459 patients who had nonfatal MI or died of CAD during the 12-year follow-up, higher plasma levels of vWF were associated with smoking and older age [76].

In another study, 73 men of age 59 ± 11 years, with stable CAD and in sinus rhythm, and 35 healthy volunteers, underwent 24-h ambulatory blood pressure monitoring [77]. Patients were divided into four groups: 1—with high pulse pressure; 2—with low pulse pressure; 3—dippers; and 4—non-dippers. The results of the study showed that all patient groups had higher plasma levels of vWF compared to controls (mean 197 ± 58 vs. 120 ± 18 IU/dL, respectively). The highest plasma levels of vWF were detected in the high pulse pressure (219 ± 58 IU/dL) and non-dipper (222 ± 55 IU/dL) groups [77]. In a study of 178 patients, 54 ± 15 years-old with arterial hypertension, who had blood pressure higher than 160/90 mmHg on two visits to a physician, and 47 normotensive healthy controls, the plasma levels of vWF were significantly higher in the hypertensive group, than in controls (mean 113 vs. 98 IU/dL, respectively) [78].

Another prospective study of 631 patients 50–75 years-old showed that increased vWF plasma levels correlated with cardiovascular and all-cause mortality, in patients with and without diabetes mellitus. The increase in plasma levels of vWF significantly correlated with older age, higher fasting glucose levels, glycated hemoglobin, body mass index, and systolic arterial pressure [79]. A study of 94 patients with non-insulin-dependent diabetes mellitus showed correlation between albuminuria and increased plasma levels of vWF [80]. In the ASCET study, age, smoking, and diabetes mellitus were associated with elevated vWF in plasma [81]. In the ENTIRE-TIMI 23 sub-study, the reduced success of thrombolysis (estimated by the thrombolysis in myocardial infarction (TIMI) flow grade and the corrected TIMI frame count) positively correlated with the plasma levels of VWF. The plasma levels of vWF in the upper quartile were associated with a higher incidence of death within 30 days, compared with the lower quartile (11.2% vs. 4.1%, respectively) [67].

Elevated plasma vWF was observed in patients with systemic inflammatory diseases. In a study of 113 patients with systemic inflammatory diseases (40 with rheumatoid arthritis, 38 with systemic scleroderma, 35 with systemic vasculitis), the plasma levels of VWF were significantly elevated, compared to 80 healthy controls [82]. Of note, in patients with chronic systemic inflammation, experiencing exacerbation in the form of serositis, plasma vWF levels, and ristocetin cofactor assay values were increased, compared to healthy controls, but were unrelated to clinical manifestations of the disease, including thrombotic complications [83].

One of the reasons for elevated plasma vWF is the polymorphism of Thr789Ala allele in the vWF gene, which is an independent predictor of CAD development [84]. Additionally, the drug-induced elevation of plasma vWF might occur due to the intake of diuretics, digoxin, heparin, and oral anticoagulants [64]. On the other hand, the administration of enoxaparin decreases plasma vWF levels, compared to heparin [67].

Plasma vWF elevation might occur due to endothelial damage. Taking into consideration that vWF is predominantly secreted by the endothelium, elevated plasma vWF might be considered to be a marker of endothelial dysfunction. Endothelial dysfunction plays a major role in the pathogenesis of atherosclerosis, increasing the risk of adverse cardiovascular events [85]. This is demonstrated effectively in a study of 50 patients with CAD who underwent coronary artery stenting with single or multiple bare-metal stents (BMS), and 8 controls with CAD who underwent only diagnostic coronary angiography [86]. The single stent implantation group comprised 25 persons of mean age 59.7 ± 9.44 years, and the multiple stent implantation group comprised 25 persons of mean age 57.5 ± 8.1 years. Blood for the measurement of vWF was taken from the coronary sinus, before and after PCI. There was no significant difference in vWF plasma levels before and after coronary angiography in the controls (123 ± 10.8 IU/dL before, 125 ± 11.6 IU/dL after). In patients with a single stent implantation, the plasma levels of vWF rose slightly from 113.6 ± 39.6 to 121.35 ± 46.63 IU/dL after stenting. In patients with multiple stent implantation, a steep rise in the plasma levels of vWF was detected, from 112.7 ± 25.16 IU/dL to 152.78 ± 41.03 IU/dL. Thus, multiple stenting significantly increased the plasma levels of vWF in coronary circulation [86]. Another study demonstrated the different influence on vWF plasma levels of BMS and drug-eluting stents (DES) [87]. In total, 16 patients with the proximal stenosis of left anterior descending artery received BMS or DES. The plasma levels of vWF were measured in the aorta and the coronary sinus immediately before and 2 h after stenting. The systemic plasma levels of vWF were measured at the baseline and 24 h after stenting. The baseline plasma levels of vWF were similar in both groups. The change in plasma levels of vWF in the coronary sinus, 2 h after stent implantation, was +20.1 ± 26.9% in the BMS group, and −5.7 ± 23.02% in the DES group. The systemic baseline plasma levels rose from 132.8 ± 58.8% to 169 ± 40.7% in the BMS group and slightly decreased from 140.6 ± 84% to 136 ± 39.5% in the DES group, 24 h after stent implantation [87].

Therefore, the elevated plasma levels of vWF do not necessarily reflect a causal relationship with the development of thrombotic complications of CAD. Rather, vWF levels rise in response to acute inflammatory stimuli and stress. In acute inflammation, vWF levels rise and fall together with C-reactive protein [88]. Different mediators of stress and inflammation influence vWF release from endothelial cells. Thus, interleukin-6 (IL-6) in a complex with the soluble IL-6 receptor, IL-8, and tumor necrosis factor-α (TNF-α), significantly stimulate the release of vWF from the endothelial Weibel–Palade bodies. IL-6 interferes in vWF cleavage by ADAMTS-13 under flow, but not static, conditions [89]. Vasopressin, which rises in response to stress stimuli, induces the release of vWF from endothelial Weibel–Palade bodies and substantially increases the plasma levels of vWF. Its analogue desmopressin (DDAVP) is used in the treatment of vWD to sustain vWF plasma levels [90]. Thus, the rise in vWF plasma levels in acute MI might reflect a reaction to ischemic injury and endothelial dysfunction, rather than a causal role in MI development.

The physiology of vWF demands high shear rates for activation and proper functioning, which future research into the role of vWF in the development of thrombotic complications of CAD should take into consideration. Preliminary data show that the shear stress-induced activation of vWF might play a role in the premature development of MI [91].

## 8. The Potential for New Treatment, Targeting von Willebrand Factor and ADAMTS-13 in Cardiovascular Diseases

Most medications, routinely used in CAD to prevent thrombotic complications, do not affect vWF and ADAMTS-13. Only heparins were shown to bind to a site on vWF that overlapped the A1 domain, thus impairing the GPIb-mediated platelet adhesion, measured by the ristocetin-cofactor activity [92]. In clinical studies, administration of enoxaparin, a low-molecular-weight heparin, was associated with a decrease in the plasma levels of vWF, MI frequency, and death in MI [67,93].

At the turn of the century, antibodies against the A1 domain of vWF, such as AJvW-2 and AJW200, were studied [94]. Despite promising preliminary results, no further studies and clinical investigations were reported. Current vWF-specific therapeutics is under development focus on aptamer antagonists of the A1 domain of vWF. The administration of the first-generation aptamer, ARC1779, to healthy volunteers, resulted in the dose- and concentration-dependent inhibition of vWF activity [95]. The ARC1779 antithrombotic effect was also studied on 36 patients who underwent carotid endartherectomy. It was shown that ARC1779 inhibits vWF activity and reduces thromboembolism in humans, but also provokes bleeding complications [96]. Recently, the development of novel aptamers was reported [97,98]. The only drug targeting the A1 domain of vWF that is approved for clinical use is ALX-0081 (caplacizumab) [47]. Caplacizumab is a humanized bivalent nanobody that specifically binds to the GPIb binding site on the A1 domain of vWF. Following remarkable results of the HERCULES trial, in which its administration resulted in a lower incidence of TTP-related death, thromboembolism and the recurrence of TTP [47], caplacizumab was approved in the European Union in 2018 and in the USA in early 2019, to treat adults with TTP. Studies of caplacizumab safety and efficacy in patients with CAD are sparse. The efficacy of caplacizumab was shown in ex vivo study of 9 patients with CAD and 11 healthy controls [99]. Caplacizumab completely inhibited platelet adhesion and aggregation, measured by ristocetin-cofactor assay, platelet function analyzer, and in flow chambers. The efficacy of caplacizumab was not influenced by antithrombotic medications, which included aspirin, clopidogrel, and heparin [99]. In a randomized, placebo-controlled phase Ib trial on 46 patients with stable CAD, undergoing PCI, administration of caplacizumab was safe, and resulted in the complete inhibition of platelet aggregation, measured by the ristocetin-cofactor assay [100]. In 2009, a phase II, randomized, open-label clinical trial was initiated in 380 high-risk patients with ACS, undergoing PCI. The objective was to compare the bleeding risk and effectiveness of caplacizumab and abciximab, a GPIIb/IIIa inhibitor [101]. As of 2020, no results of the study are published.

Recently, in a study on mice, treatment with recombinant human ADAMTS-13 was shown to decrease coronary vascular dysfunction and improve cardiac remodeling after left ventricular pressure overload [102]. Another study on mice showed that the administration of recombinant human ADAMTS-13 resulted in a reduction in infarct size, the neutrophil infiltration of ischemic myocardium, and lower troponin-I release [32]. Currently, there are no therapeutics targeting vWF and ADAMTS-13 approved for the treatment of cardiovascular diseases.

## 9. Conclusions

vWF plays an important role in cardiovascular disease. The deficiency of HMWM of vWF in valvular heart disease and obstructive hypertrophic cardiomyopathy manifests with gastrointestinal, skin or mucosal bleeding. The bleeding might also complicate the surgical treatment of such patients. In CAD, the involvement of vWF is more controversial. Though abundant data show that the plasma levels of vWF are increased and ADAMTS-13 levels are decreased in CAD, especially in MI, this does not necessarily reflect a causal relationship between elevated plasma vWF and MI. The existing data show that vWF levels rise in response to stress and acute inflammatory stimuli, which occur in acute MI. Rather than focusing on the measurement of plasma levels of vWF and the evaluation of vWF activity in static or low shear rate conditions, future research should concentrate on physiologically relevant studies of vWF functions under high shear rates. This might pave a way to new approaches to measure vWF function, which eventually might turn vWF into a treatment target or an important tool for diagnostics and risk assessment in cardiovascular disease.

## Figures and Tables

**Figure 1 ijms-21-07804-f001:**
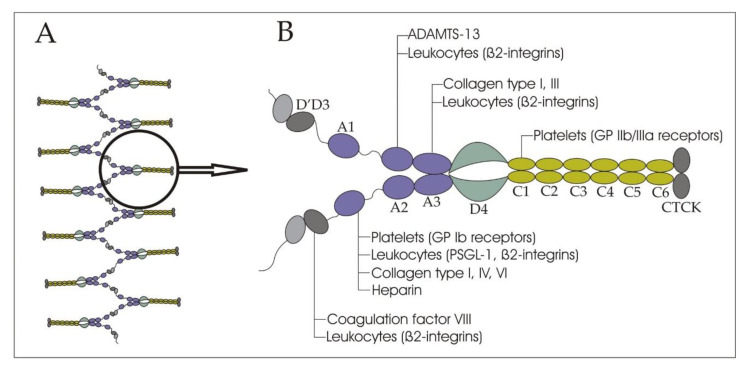
(**A**) A schematic representation of a von Willebrand factor thread; and (**B**) a von Willebrand factor dimer. The binding sites for ADAMTS-13, platelets, leukocytes, collagen, and heparin are indicated.

**Figure 2 ijms-21-07804-f002:**
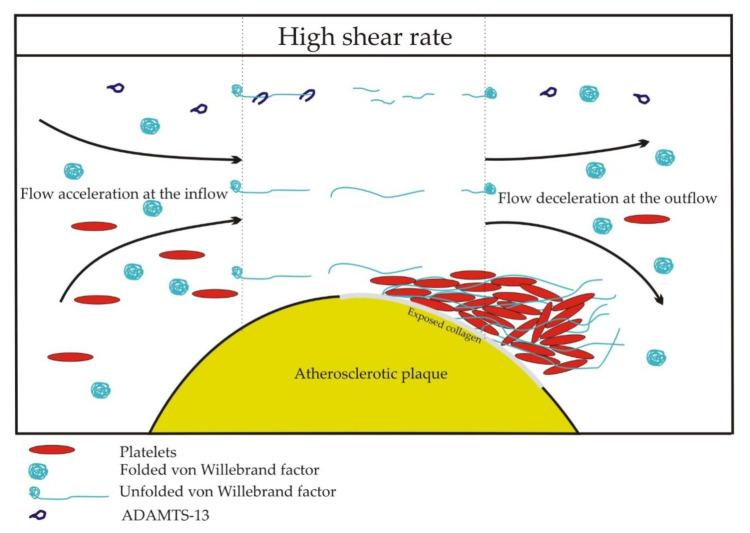
Shear rate-induced activation of the von Willebrand factor at the site of atherosclerotic narrowing of a vessel.

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
