# Peer review of "Shear Stress-Induced Activation of von Willebrand Factor and Cardiovascular Pathology"

_ijms, 2020, doi:10.3390/ijms21207804_

Round 1

Reviewer 1 Report

Authors review

Authors made an effort to cover many aspects of vWF physiology and pathophysiology. I have no major criticism regarding the manuscript. I only would like to highlight some aspects which could have been elaborated on.

Speaking of vWF multimerization and shear stress it is worth to mention that vWF can self-associate to form larger complexes at high shear (von Willebrand factor self-association is regulated by the shear-dependent unfolding of the A2 domain. Zhang C, Kelkar A, Neelamegham S.Blood Adv. 2019 Apr 9;3(7):957-968) Therefore high shear is able not only to increase platelet binding by unfolding of vWF but also by facilitating of formation of large multimers.

Authors only briefly mention the presence of the platelet pool of vWF. Its role is rather elusive but there is one excellent publication showing that this pool may play an important role in ischemic stroke (Blood 2015 Oct 1;126(14):1715-22 Platelet-derived VWF is not essential for normal thrombosis and hemostasis but fosters ischemic stroke injury in mice)

Authors describe clinical data regarding increased production of vWf in metabolic and inflammatory diseases. There are some studies in animal models showing that increased production of vWf by inflamed endothelium can perpetuate the inflammatory processes of vascular wall by facilitating of recruitment of platelets and leukocytes to vascular wall.

Minor issues:

I would suggest to replace the term “cut vessels” with “injured vessels”

Author Response

Reply to the reviewer 1.

Point 1: Speaking of vWF multimerization and shear stress it is worth to mention that vWF can self-associate to form larger complexes at high shear (von Willebrand factor self-association is regulated by the shear-dependent unfolding of the A2 domain. Zhang C, Kelkar A, Neelamegham S.Blood Adv. 2019 Apr 9;3(7):957-968) Therefore high shear is able not only to increase platelet binding by unfolding of vWF but also by facilitating of formation of large multimers.

Response 1:

The following text was added (lines 105-114): “Unfolded by high shear rates, vWF threads can self-associate and form long strands and web-like structures, further stimulating platelet adhesion. The ability of unfolded vWF to self-associate under high-shear rates was studied in vitro in endothelialized microvessels. Smaller vessels, turns, bifurcations, flow acceleration predisposed to thickening and elongation of vWF strands. In regions with complex flow, vWF strands tended to form web-like structures. At the same time, at sites of low shear rates vWF remained in a globular conformation and no vWF strands were visible [19]. Schneider et al. also reported spider web-like network formation by vWF under high shear rates [15]. Zhang et al. demonstrated that vWF association was dependent on the A2 domain under shear rate of 9600s-1. Hence, the A2 domain may have an overlapping role both in proteolysis and self-association of vWF [20]”.

Point 2: Authors only briefly mention the presence of the platelet pool of vWF. Its role is rather elusive but there is one excellent publication showing that this pool may play an important role in ischemic stroke (Blood 2015 Oct 1;126(14):1715-22 Platelet-derived VWF is not essential for normal thrombosis and hemostasis but fosters ischemic stroke injury in mice).

Response 2:

The following text was added (lines 115-123): "The role of platelet vWF in hemostasis is not clearly established, and studies here are quite sparse. A study on vWF-knockout mice showed that replenishing of vWF in platelets after bone marrow transplantation from wild-type mice resulted in partial correction of bleeding time and reduction of blood loss volume [21]. Recently, a study showed that chimeric mice with only platelet vWF had bleeding time comparable to that of vWF-knockout mice. Platelet vWF did not contribute to thrombus formation in a model of carotid artery injury. On the other hand, after ischemic brain injury the mice with only platelet vWF, and without plasma vWF, had cerebral infarction size similar to the wild-type mice. Platelet vWF mediated ischemic brain injury in a GPIb-dependent mechanism and significantly contributed to inracerebral thrombosis formation [22]".

Point 3: Authors describe clinical data regarding increased production of vWf in metabolic and inflammatory diseases. There are some studies in animal models showing that increased production of vWf by inflamed endothelium can perpetuate the inflammatory processes of vascular wall by facilitating of recruitment of platelets and leukocytes to vascular wall.

Response 3:

The following text was added (lines 152-167): "The growing evidence shows that vWF is an important mediator of vascular inflammation. A murine model of acute peritonitis showed that vWF plays an important role in leukocyte recruitment and extravasation into the site of inflammation. Notably, leukocyte extravasation requires vWF-platelet binding, as inhibition of platelet GPIb receptors prevented vWF-mediated leukocyte extravasation [28]. A murine model of immune complex-mediated vasculitis and irritant contact dermatitis showed that the interference with vWF activity with vWF-blocking antibody led to a substantial decrease in vWF-mediated leukocyte recruitment and reduction of cutaneous inflammatory response. In this study, vWF-blocking antibody did not interfere with the GPIb domain [29]. Another murine model with the same pathology but with the use of the antibody targeting the A1 domain showed the comparable results [30]. A study comparing features of acute inflammation after focal cerebral ischemia in vWF-deficient and ADAMTS-13-deficient mice, showed that the vWF-deficient mice had a smaller injury size, reduced cytokine release and neutrophil infiltration of the injury site, compared to the ADAMTS-13 deficient mice [31]. A myocardial ischemia/reperfusion study in ADAMTS-13-knockout mice showed that infusion of recombinant ADAMTS-13 substantially reduced myocardial apoptosis, troponin-I release, and resulted in the 9-fold reduction in number of neutrophils infiltrating the ischemic zone [32]".

Minor issues: I would suggest to replace the term “cut vessels” with “injured vessels”

Response: the term ‘cut vessels’ was replaced with ‘injured vessels’ throughout the text.

Reviewer 2 Report

This review article describes the relationship between shear stress-induced activation of von Willebrand factor and cardiovascular pathology, as well as the possible diagnostic and treatment approaches. However, this paper suffers from some drawbacks.

Major points:

First, some state-of-the-art studies in this area are absent.

Second, some statements lack focus and are wordy.

Minor points:

Title and text. A hyphen should be placed between stress and induced.

L127. thyronine should be tyrosine.

L422. Adenoside diphosphate should be Adenosine diphosphate.

Author Response

Major points:

 Point 1: First, some state-of-the-art studies in this area are absent.

Response 1: We have added the following substantial data.

The data on the self-association of vWF (lines 105-114): “Unfolded by high shear rates, vWF threads can self-associate and form long strands and web-like structures, further stimulating platelet adhesion. The ability of unfolded vWF to self-associate under high-shear rates was studied in vitro in endothelialized microvessels. Smaller vessels, turns, bifurcations, flow acceleration predisposed to thickening and elongation of vWF strands. In regions with complex flow, vWF strands tended to form web-like structures. At the same time, at sites of low shear rates vWF remained in a globular conformation and no vWF strands were visible [19]. Schneider et al. also reported spider web-like network formation by vWF under high shear rates [15]. Zhang et al. demonstrated that vWF association was dependent on the A2 domain under shear rate of 9600s-1. Hence, the A2 domain may have an overlapping role both in proteolysis and self-association of vWF [20]”.

The data on the role of platelet vWF in hemostasis (lines 115-123): “The role of platelet vWF in hemostasis is not clearly established, and studies here are quite sparse. A study on vWF-knockout mice showed that replenishing of vWF in platelets after bone marrow transplantation from wild-type mice resulted in partial correction of bleeding time and reduction of blood loss volume [21]. Recently, a study showed that chimeric mice with only platelet vWF had bleeding time comparable to that of vWF-knockout mice. Platelet vWF did not contribute to thrombus formation in a model of carotid artery injury. On the other hand, after ischemic brain injury the mice with only platelet vWF, and without plasma vWF, had cerebral infarction size similar to the wild-type mice. Platelet vWF mediated ischemic brain injury in a GPIb-dependent mechanism and significantly contributed to inracerebral thrombosis formation [22]”.

The data on the role of vWF as a mediator of vascular inflammation (lines 152-167): “The growing evidence shows that vWF is an important mediator of vascular inflammation. A murine model of acute peritonitis showed that vWF plays an important role in leukocyte recruitment and extravasation into the site of inflammation. Notably, leukocyte extravasation requires vWF-platelet binding, as inhibition of platelet GPIb receptors prevented vWF-mediated leukocyte extravasation [28]. A murine model of immune complex-mediated vasculitis and irritant contact dermatitis showed that the interference with vWF activity with vWF-blocking antibody led to a substantial decrease in vWF-mediated leukocyte recruitment and reduction of cutaneous inflammatory response. In this study, vWF-blocking antibody did not interfere with the GPIb domain [29]. Another murine model with the same pathology but with the use of the antibody targeting the A1 domain showed the comparable results [30]. A study comparing features of acute inflammation after focal cerebral ischemia in vWF-deficient and ADAMTS-13-deficient mice, showed that the vWF-deficient mice had a smaller injury size, reduced cytokine release and neutrophil infiltration of the injury site, compared to the ADAMTS-13 deficient mice [31]. A myocardial ischemia/reperfusion study in ADAMTS-13-knockout mice showed that infusion of recombinant ADAMTS-13 substantially reduced myocardial apoptosis, troponin-I release, and resulted in the 9-fold reduction in number of neutrophils infiltrating the ischemic zone [32]”.

The text describing vWF multimers analysis by electrophoresis was re-written (lines 193-200): “vWF multimer analysis by the electrophoresis of vWF on agarose gels remain the main assay of qualitative deficiency of vWF. The assay has excellent sensitivity for the loss of HMWM of vWF. The inclusion of the luminescent methods gives it the power to detect the presence of vWF in samples with a concentration of less than 1 IU/dL. This allows differentiation between homozygous type 2 and type 3 vWD. vWF multimer analysis is crucially important for subtyping of type 2 vWD [37]. It is also very useful in diagnostic of HMWM of vWF deficiency in acquired von Willebrand syndrome due to heart valve disease [38]. However, the assay places high demands on the technical competence of the laboratory personnel and is unsuitable for standardization [39]”

The description of thrombotic thrombocytopenic purpura was enhanced (lines 244-261): “TTP can develop either due to genetic anomalies (congenital TTP) or production of autoantibodies to ADAMTS-13 (acquired TTP), which interfere with its function or enhance clearance. The determination of ADAMTS-13 function is essential in TTP diagnosis. TTP usually manifests when ADAMTS-13 levels fall below 10%, which is below the detection limit of many assays. As soon as ADAMTS-13 activity becomes detectable (i.e. higher than 10%), TTP symptoms resolve. Differentiation between congenital and acquired TTP is vitally important, as treatment of these life-threatening conditions significantly differ. The diagnosis of congenital TTP is built on the detection of pronounced fall in ADAMTS-13 activity, ruling out autoantibodies to ADAMTS-13 and analyzing the ADAMTS-13 gene. The diagnosis of acquired TTP requires the lack of ADAMTS-13 activity and the detection of autoantibodies to ADAMTS-13. Patients with congenital TTP respond well to the plasma exchange therapy, which improves survival rates from 10% to 80-90%. During the plasma exchange, ultra-large multimers of vWF, immune complexes, and autoantibodies are removed. Infusions of donor plasma with normal content of ADAMTS-13 are also effective, and used in combination with the plasma exchange therapy. Patients with acquired TTP respond to normal plasma infusions poorly. Suppression of autoantibody production with corticosteroids, destruction of CD20-posititve B-lymphocytes with rituximab, or the use of other immunosuppressive agents is considered to be effective in acquired TTP. Splenectomy is effective, but reserved for refractory cases of acquired TTP [46]”.

The data on acquired von Willebrand syndrome in heart valve disease was added (lines 267-276): “association between aortic stenosis and cryptogenic gastrointestinal bleeding was first described in 1958 in a letter by E.C. Heyde [48]. In 1992, Warkentin et al. were the first to suggest a link between the loss of HMWM of vWF that develops due to aortic stenosis or hypertrophic cardiomyopathy and bleeding from gastrointestinal angiodysplasia [49]. In 2002, Warkentin et al. was also the first to provide a rationale for this association. They reported 2 cases of severe aortic stenosis with concomitant gastrointestinal bleeding that ceased after successful replacement of the aortic valve. Notably, prior to operation the platelet count, activated partial-thromboplastin time, the plasma level of FVIII, the plasma levels of vWF and ristocetin cofactor assay values were normal, whereas the loss of HMWM of vWF was severe. After the valve replacement the percentage of HMWM recovered, and remained normal during 10-year follow up [50]”;

 and lines 284-290: “Panzer et al. reported a study of 47 patients with severe aortic stenosis who were subjected to aortic valve replacement. HMWM of vWF were depleted in all patients before valve replacement and became normal in most of the patients. PFA-100 closure time in collagen-ADP cartridges were significantly prolonged at the baseline, and normalized after the treatment. Using the cone and plate analyzer, the authors showed that the loss of HMWM of vWF affects platelet adhesion and, to a larger extent, ADP-induced platelet aggregation [52]”.

Point 2: Second, some statements lack focus and are wordy.

Response 2: We excluded introductory and other unnecessary parts.

The following text was excluded (lines 58-61): “The regulation of vWF size in circulation is vitally important, as the persisting presence of ultra-large multimers in the bloodstream leads to spontaneous thrombus formation with concomitant bleeding due to the depletion of circulating platelets, known as thrombotic thrombocytopenic purpura (TTP)”. Reason: there is a detailed description of TTP in the section 4 ‘Diseases, associated with Von Willebrand factor and ADAMTS-13 dysfunction’.

 The following text was excluded (lines 124-128): “Unfolded vWF multimers participate in hemostasis in the following way. Through the A1 and A3 domains, they attach to the collagen of the intercellular matrix of a damaged vessel wall, forming long threads. These threads detain platelets through the interaction of the A1 domain with GPIb receptors, and mediate platelet adhesion to a damaged vessel wall. vWF can also increase platelet adhesion through the interaction of the C1 domain and GP IIb/IIIa receptors of activated platelets”. Reason: unnecessary description, as these data are provided in other paragraphs.

The following text was excluded (lines 238-240): “Normally, ultra-large multimers are rapidly cleaved by ADAMTS-13 upon secretion from the endothelium into the pool of HMWM and smaller less-active multimers [10]. When ADAMTS-13 is deficient, circulating ultra-large multimers of vWF”. Reason: repetitive.

The following text was excluded (lines 298-299): “Platelet function analyzer-100 (PFA-100), which measures platelet function under conditions of high shear rates”. Reason: it was stated earlier (lines 207-217).

The following text was excluded (lines 341-356): “Under normal conditions, the hemostatic activity of vWF is well-balanced by ADAMTS-13. In coronary artery disease (CAD), the balance is lost. Atherosclerotic plaques narrow vessel lumen and create increased shear rates, which unfold vWF multimers. Collagen threads, which are exposed in the case of plaque rupture, serve as a binding substrate, to which unfolded vWF multimers adhere rapidly. Then, platelets from circulation are recruited through the interaction of GPIb receptors with the A1 domain of vWF. Activated platelets release bioactive agents, the concentration of which is increased by more than 50 times at a site of local thrombosis. Platelets release even more vWF from α-granules, which further contribute to thrombus growth. Then, thrombus stabilizes and occludes a vessel [62]. Unfolded vWF multimers may also play a role in leukocyte recruitment. Leukocytes adhere poorly to the endothelium under high shear rates. Model studies showed that vWF facilitates leukocyte recruitment under low and high shear rates, but under high shear rates, the more complex interaction, involving platelets, is required. Unfolded vWF multimers present binding sites for PSGL-1 and β2-integrins of leukocytes. Platelets, adhered to unfolded vWF multimers, express P-selectin at much higher levels than endothelial cells and significantly intensify leukocyte recruitment. Therefore, unfolded by high shear rates, vWF may not only facilitate thrombosis development, but also recruit leukocytes to atherosclerotic plaques [63]”.  Reason: repetitive. The data on leukocytes is important, but in the revised manuscript it is provided in the pathophisiologic section (lines 152-167).

The following text was excluded (line 412): “The depletion of ADAMTS-13 is thought to occur due to its response to increased vWF activity in acute MI”. It was already mentioned in the line 146.

We also made minor corrections to exclude extra wording, such as exclusion of unnecessary ‘Of note’, abbreviation of ‘von Willebrand disease’ to ‘vWD’, etc.

Minor

Point 1: Title and text. A hyphen should be placed between stress and induced.

L127. thyronine should be tyrosine.

L422. Adenoside diphosphate should be Adenosine diphosphate.

Response 1: A hyphen was placed between ‘stress’ and ‘induced’ in the title and in line 530. ‘Thyronine’ was replaced with ‘tyrosine’ (line 150). A mistake in ‘Adenoside’ was corrected (line 210). We have also made other minor corrections throughout the text.

Reviewer 3 Report

The review is comprehensive. However, I feel that it could be organized a bit different:

start with the diagnostic tools (and there I believe the gold standard still is electrophoresis, here Budde should be cited!), and make clear, that the PFA-100 was specially developed to rapidly diagnose vWD, even though it is a rather crude estimation.

In the introduction to acquired vW I think the key reports by TE Warketin need to be cited. As well as the observations in aortic stenosis by S Panzer.

Further, I miss the differentiation between acquired and inherited ADAMTS-13 deficiency (and therapeutic consequences).

In the listing and reviewing  CAD and levels of vW I believe that most authors did not disclose blood groups. You correctly mention the influence of blood group 0, but you may critically view the cited papers in this context. Thus, the influence of Blood groups shall be mentioned already in the diagnostic tools.

Author Response

Point 1: Extensive editing of English language and style is required.

Response 1: we believe that you might have received the copy of the manuscript before it was proofread. We applied for the English editing by the MDPI soon after the submission. Please find the sertificate in the attached file.

If you still feel that English of the paper shall be amended please let us know.

Point 2: start with the diagnostic tools (and there I believe the gold standard still is electrophoresis, here Budde should be cited!), and make clear, that the PFA-100 was specially developed to rapidly diagnose vWD, even though it is a rather crude estimation.

Response 2: we moved the diagnostic section to the beginning, but placed it after the pathophisiologic section ‘Structure and functions of VWF in bloodstream’. This way it is clearer why different diagnostic tools are used and where the differences in results arise from.

The lines about vWF mutimer analysis were re-written (lines 193-200): "vWF multimer analysis by the electrophoresis of vWF on agarose gels remain the main assay of qualitative deficiency of vWF. The assay has excellent sensitivity for the loss of HMWM of vWF. The inclusion of the luminescent methods gives it the power to detect the presence of vWF in samples with a concentration of less than 1 IU/dL. This allows differentiation between homozygous type 2 and type 3 vWD. vWF multimer analysis is crucially important for subtyping of type 2 vWD [37]. It is also very useful in diagnostic of HMWM of vWF deficiency in acquired von Willebrand syndrome due to heart valve disease [38]. However, the assay places high demands on the technical competence of the laboratory personnel and is unsuitable for standardization [39]"

Budde was cited.

This text was added to the lines 203-207: “Thus, the broad panel of tests is needed to diagnose vWD and its type. To simplify testing (e.g. in screening for bleeding disorders or in the emergency setting), a screening tool that can quickly exclude platelet defects and vWD was introduced. The Platelet Function Analyzer-100 (PFA-100) is an easy to use device, which utilizes whole blood and requires only 5 minutes to complete a test [41]”.

The line 207 was re-written: “Currently, this is the only commercially available system that evaluates primary hemostasis under high shear rate conditions”.

This text was added to the lines 215-217: “Its sensitivity varies depending on the severity of vWF deficiency and platelet defects, from moderate in mild cases to 100% sensitivity in the case of complete absence of vWF in blood”.

Point 3: In the introduction to acquired vW I think the key reports by TE Warketin need to be cited. As well as the observations in aortic stenosis by S Panzer.

Response 3:

The following text was added to the lines 266-275: "The association between aortic stenosis and cryptogenic gastrointestinal bleeding was first described in 1958 in a letter by E. Heide, MD [48]. In 1992, Warkentin et al. were the first to suggest a link between the loss of HMWM of vWF that develops due to aortic stenosis or hypertrophic cardiomyopathy and bleeding from gastrointestinal angiodysplasia [49]. In 2002, Warkentin et al. was also the first to provide a rationale for this association. They reported 2 cases of severe aortic stenosis with concomitant gastrointestinal bleeding that ceased after successful replacement of the aortic valve. Notably, prior to operation the platelet count, activated partial-thromboplastin time, the plasma level of FVIII, the plasma levels of vWF and ristocetin cofactor assay values were normal, whereas the loss of HMWM of vWF was severe. After the valve replacement the percentage of HMWM recovered, and remained normal during 10-year follow up [50]".

The following text was added to the lines 283-288: ”Panzer et al. reported a study of 47 patients with severe aortic stenosis who were subjected to aortic valve replacement. HMWM of vWF were depleted in all patients before valve replacement and became normal in most of the patients. PFA-100 closure time in collagen-ADP cartridges were significantly prolonged at the baseline, and normalized after the treatment. Using the cone and plate analyzer, the authors showed that the loss of HMWM of vWF affects platelet adhesion and, to a larger extent, ADP-induced platelet aggregation [52]”.

Point 4: Further, I miss the differentiation between acquired and inherited ADAMTS-13 deficiency (and therapeutic consequences).

Response 4:

The following text was added (lines 243-260): “TTP can develop either due to genetic anomalies (congenital TTP) or production of autoantibodies to ADAMTS-13 (acquired TTP), which interfere with its function or enhance clearance. The determination of ADAMTS-13 function is essential in TTP diagnosis. TTP usually manifests when ADAMTS-13 levels fall below 10%, which is below the detection limit of many assays. As soon as ADAMTS-13 activity becomes detectable (i.e. higher than 10%), TTP symptoms resolve. Differentiation between congenital and acquired TTP is vitally important, as treatment of these life-threatening conditions significantly differ. The diagnosis of congenital TTP is built on the detection of pronounced fall in ADAMTS-13 activity, ruling out autoantibodies to ADAMTS-13 and analyzing the ADAMTS-13 gene. The diagnosis of acquired TTP requires the lack of ADAMTS-13 activity and the detection of autoantibodies to ADAMTS-13. Patients with congenital TTP respond well to the plasma exchange therapy, which improves survival rates from 10% to 80-90%. During the plasma exchange, ultra-large multimers of vWF, immune complexes, and autoantibodies are removed. Infusions of donor plasma with normal content of ADAMTS-13 are also effective, and used in combination with the plasma exchange therapy. Patients with acquired TTP respond to normal plasma infusions poorly. Suppression of autoantibody production with corticosteroids, destruction of CD20-posititve B-lymphocytes with rituximab, or the use of other immunosuppressive agents is considered to be effective in acquired TTP. Splenectomy is effective, but reserved for refractory cases of acquired TTP [46]”.

Point 5: In the listing and reviewing CAD and levels of vW I believe that most authors did not disclose blood groups. You correctly mention the influence of blood group 0, but you may critically view the cited papers in this context. Thus, the influence of Blood groups shall be mentioned already in the diagnostic tools.

Response 5: The paragraph describing blood groups was moved to the diagnostic section (now lines 177-182).

The following text was added (lines 175-177): “The plasma levels of vWF depend on the blood group of a person. Variations of the plasma levels of vWF between blood groups may complicate distinguishing healthy persons with low vWF from mild cases of vWD”.

Minor changes:

The term ‘von Willebrand disease’ was abbreviated to ‘vWD’ throughout the text.

Round 2

Reviewer 2 Report

The revised manuscript is now acceptable for publication. 

Author Response

Thank you for the valuable review and commentary!

Reviewer 3 Report

The authors have adjusted the manuscript along the suggested lines.

Author Response

(The authors gave the same response as above.)
